# Characterization of methane microseepage from natural gas reservoirs in mild climate: A case study of Xinchang gas field

**Haoqi Wang** [1]*, **Maoyi Tang**[1], **Huimin Yun**[2]*, **Zhourong Ke**[1], **Guojian Wang**[3], **Chongxin Huang**[1], **Weiwei Ji**[1], **Caixin Pu**[1], **Junhong Tang**[1]

**1** College of Material and Environmental Engineering, Hangzhou Dianzi University, Zhejiang, China, **2** College of Chemical Engineering, Beijing University of Chemical Technology, Beijing, China, **3** Wuxi Research Institute of Petroleum Geology, Research Institute of Petroleum Exploration and Production, Sinopec, Jiangsu, China

* chris.haoqi.wang@gmail.com (HW); yunhuimin@mail.buct.edu.cn (HY)

## Abstract

Methane microseepage from oil and gas fields significantly contributes to atmospheric methane level, making it a critical factor in global climate change. Therefore, accurate monitoring of surface flux and investigating migration mechanism are pivotal to evaluating and mitigating the impact of methane microseepage. In this study, methane microseepage from natural gas reservoirs in a mild climate was investigated, using Xinchang gas field as a case study. Soil samples were collected to analyze geochemical anomalies of acid-hydrolyzed hydrocarbons (AHH) and altered carbonates (AC). Surface methane flux from natural gas reservoirs were monitored, using a greenhouse gas analyzer and static gas collection chambers. Methane release patterns and migration mechanism were then discussed. Headspace and soil gas samples were collected to determine the hydrocarbon composition and carbon isotope profile. The results indicate that surface methane flux in Xinchang gas field is weak, exhibiting three release patterns: continuous, episodic, and flat. Spiked anomalies of AHH and AC co-exist in the test area, suggesting methane migration from reservoirs to surface. Hydrocarbon composition and carbon isotope profile in headspace and soil gas samples confirm thermogenic origin of methane. These findings offer new insights into the behavior of methane microseepage from natural gas reservoirs in mild climate. It is also suggested that close monitoring and stringent regulation of methane microseepage, as well as continuous investigation on factors affecting this phenomenon, are essential to the management of geological methane emissions. The conclusions of this work align with previous studies and are applicable to managing methane microseepage from oil and gas reservoirs in a wider scope.

## 1. Introduction

Methane is a potent and common greenhouse gas (GHG), which may originate from anthropogenic and natural sources. A critical fraction of methane emissions from natural sources comes from geological methane, to which methane microseepage from oil and gas reservoirs

**Data availability statement:** All relevant data are within the manuscript and its Supporting Information files.

**Funding:** This work is financially supported by the Natural Science Foundation of China (NSFC Grants No. U2003101, 41872126, and 41373121), Hangzhou Dianzi University (Grants No. KYS205622065), and China Postdoctoral Science Foundation. The funders had no role in study design, data collection and analysis, decision to publish, or preparation of the manuscript.

**Competing interests:** The authors have declared that no competing interests exist.

contributes the greatest [1,2]. Methane microseepage manifests as low and invisible, but persistent and pervasive loss of methane in sedimentary basins [3,4]. Especially in geological structures bearing rich oil and gas resources, methane and light hydrocarbons can break through sealing and migrate to surface, driven by such forces as diffusion and convection [5]. It is found that methane microseepage is observed in more than half of basins bearing oil and gas resources with data available [6].

Methane and other hydrocarbons reaching earth surface may be captured by soil and water in adsorbed or free forms, or consumed by microbial activities, resulting in detectable geochemical anomalies, which is widely exploited for oil and gas exploration [7,8]. To further complicate the matter, while anaerobic decomposition of organic matters in soil generates methane, which needs to be distinguished from methane of fossil origin via analyzing features such as hydrocarbon composition and carbon isotope [9], methanotrophic microbial activities and atmospheric oxidation consume methane near earth surface, which is known to be strongly affected by climate conditions [1,10]. Therefore, in mild climate, the interference of rich microbial presence on soil methane balance makes it difficult to accurately determine methane microseepage flux.

If not otherwise captured or consumed, methane migrating from reservoirs manifests as surface flux that can be monitored. In petroleum geology, the flux chamber method is used to capture hydrocarbons [11], making it possible to measure methane concentration by greenhouse gas analyzers. On realization that methane microseepage is indispensable to effective GHG management, monitoring its flux and evaluating its impact on atmospheric methane levels have been carried out globally since the 2000s [3,12,13]. However, to the best of our knowledge, research on methane microseepage in Asia is scarce, except for several studies conducted in oil and gas fields in Tarim Basin in China [9,14] and in mud volcano regions in eastern Azerbaijan [15]; Particularly, no such work has focused on methane microseepage from hydrocarbon reservoirs in mild climate.

Xinchang gas field, located in the Sichuan Basin in southwest China, offers a unique case study to investigate the behavior of methane microseepage from natural gas reservoirs in mild climate. The gas field is a large-scale compound natural gas field located in the Chuanxi Depression in the west of Sichuan Basin (Fig 1). With trapping area of 900 km², depth of 4900 m, and high pressure, the field is rich in hydrocarbon resources, multi-layered, and stacked with various types of reservoirs [16–19]. Named after Xinchang tectonic zone, the field is composed of anticlines extending northeast. The field has an average altitude of about 500 m, high in the west and low in the east, steep in the north and slow in the south, characterized by shallow hilly landforms and flat terrain. The climate in the area is generally mild and humid, with summer of high temperature and abundant rainfall and winter of relatively lower temperature and less rain. The annual average temperature is in the range of 15–17 °C, and the daily average temperature is above 0 °C throughout the year.

Xinchang gas field is composed of a continental clastic rock sedimentary system, which is superimposed by three layers of Mesozoic reservoirs spanning over 5 km vertically: The shallow layer is tectonic-lithologic, with one Cretaceous and two Jurassic reservoirs, the middle layer is tectonic-diagenetic, with three Jurassic reservoirs, and the deep layer is tectonic-fractured, with four Triassic reservoirs [16,18].

The climate in the gas field area is warm and humid, giving rise to rich vegetation and high levels of microbial activity, which directly impact biogenic methane flows. In this work, soil samples were collected in and out of the gas field area to investigate geochemical anomalies of acid-hydrolyzed hydrocarbons and altered carbonates, which serve as useful precursors of hydrocarbon migration from reservoir to surface. At test sites in the field area where such anomalies were detected, surface methane flux from natural gas reservoirs were then monitored by the combined use of static gas collection chamber and greenhouse gas analyzer, and methane release

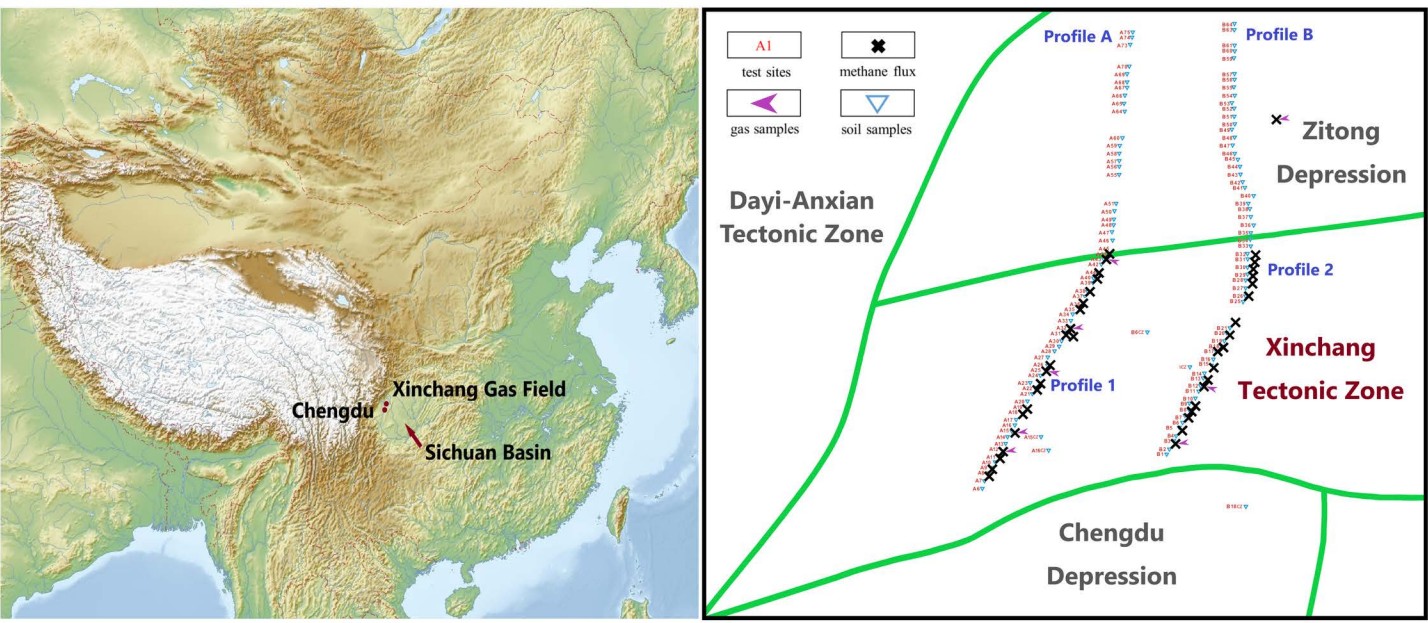

**Fig 1. Location of Xinchang gas field and distribution of test sites.**

patterns and migration mechanism were discussed. Headspace and soil gas samples were further collected where methane flux was high, for analysis of methane and heavier hydrocarbon concentrations and carbon isotopic profiles, which are crucial indicators to the hydrocarbon origin. The combination of these analyses can provide persuasive evidence for the existence of methane microseepage in the gas field and critical information on its characteristics.

## 2. Methods

### 2.1. Field work

Hydrocarbons leaking from gas reservoirs will form geochemical anomalies near earth surface, which can then be detected [20]. Based on petroleum geological data, near-surface geochemical exploration was carried out in Xinchang gas field, to collect samples of solids and gases in soil and monitor surface methane flux (Fig 1). The field work was conducted on public lands outside the natural gas mining area. Therefore, no permits were required.

 **2.1.1. Soil samples.** Soil samples were collected at depth of 1.5 m along Profiles A and B to analyze the composition of acid-hydrolyzed hydrocarbons (AHH) and the concentration of altered carbonates (AC), shown as blue triangles in Fig 2. Gas field areas where anomalies of AHH and AC were detected (detailed in the Results and Discussions Section) were denoted as Profiles 1 and 2, where surface methane flux measurement and soil gas sampling were subsequently performed.

 **2.1.2. Methane flux measurement.** Measurement of surface methane flux was conducted in November and December 2020 at 45 test sites (shown as black crosses in Fig 1), set up with intervals of 500 – 1000 m along Profiles 1 and 2. Methane flux measurement took place after the lab results for the soil samples were obtained, so the methane flux test sites were denoted differently (B2-35 for Profile 1 and C63-88 for Profile 2, both from north to south). As shown in Fig 2, surface methane flux was collected in the static gas collection chamber and then measured by a portable greenhouse gas analyzer (model UPGA-9150011) connected to the

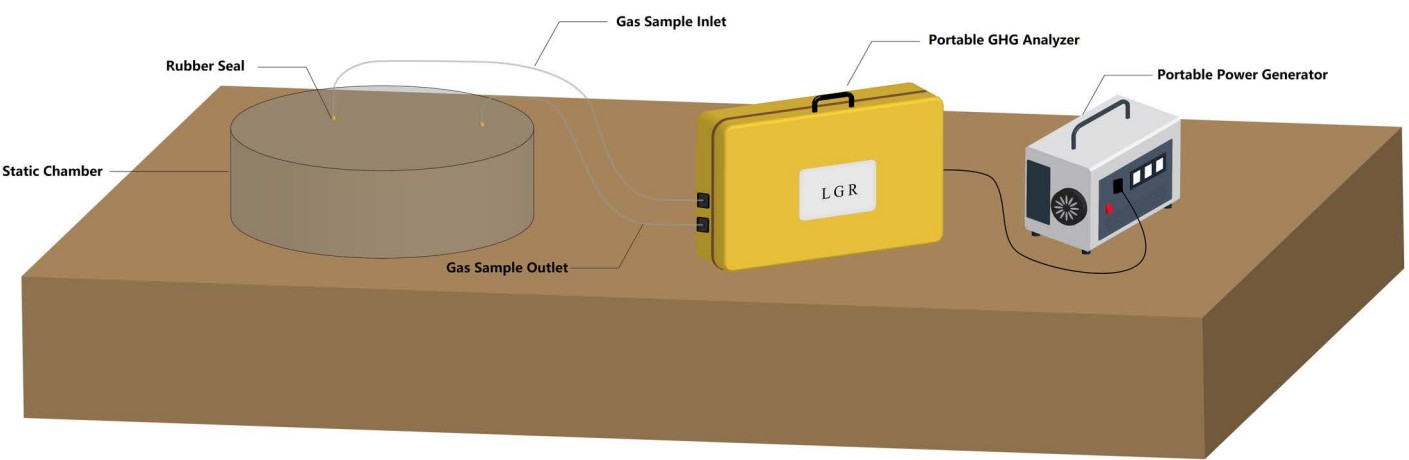

**Fig 2. Measurement of methane flux using gas analyzer and static gas collection chamber.**

chamber [21]. The collection chambers were made of polymethyl methacrylate, cylindrical with an inner diameter of 20 cm and height of 11 cm. The greenhouse gas analyzer has detection limit of 5 ppb and 1-σ precision of 0.6 ppb for methane. In the flux measurement procedure, the collection chamber was first inserted into and sealed by the soil. The gas analyzer was first calibrated by measuring the background methane concentration in the atmosphere and in the chamber. After 24 hours, the methane concentration in the chamber was again measured.

**2.1.3. Headspace and soil gas samples.** Headspace and soil gas samples were collected at surface methane flux test sites with high measurements. Headspace gas refers to hydrocarbons adsorbed by the surface of soil and rock particles [22]. Samples were collected by polyethylene gas sampling tubes at 0.3 m, 0.5 m, 0.8 m, 1.0 m, and 1.2 m underground. Soil gas refers to hydrocarbons present in the free form in voids among soil and rock particles [23]. Samples were collected at 1.5 m underground using a drill with a gas collection chamber on the tip - as the drill squeezed the soil particles the designated depth, gases in the soil were forced out and then captured directly by the gas collection chamber. All gas samples were stored in saturated brine for laboratory analysis.

## 2.2. Laboratory analysis

The analysis of acid-hydrolyzed hydrocarbon compositions in soil samples was based on the Part 5 of the Chinese National Standards GB/T 29173–2012 [24]. Each sample was dried in a cool and ventilated room, crushed, screened, and then placed into a flask. The flask was then dosed with potassium hydroxide and hydrochloric and heated by water bath for degassing. Extracted gas was then analyzed in gas chromatograph (GC) for determination of hydrocarbon composition. The analysis of altered carbonate concentrations in soil samples was based on the Part 10 of GB/T 29173–2012 [24]. Each sample was fully oxidized in a furnace at constant temperature of 500 °C for 1 h. After oxidation and cooling, the sample was then decomposed at constant temperature of 600 °C, and the concentration of carbon dioxide in the decomposition gas was measured by GC.

The analysis of hydrocarbon compositions in headspace and soil gas samples, including methane, ethane, ethylene, propane, propylene, n/i-butane, and n/i-pentane, was based on the Part 9 of GB/T 29173–2012 [24]. GC was performed by Agilent-6890N system, in $Al_2O_3$ columns at 120°C. The analysis of carbon isotopic composition of methane and carbon dioxide in gas samples was based on the Chinese National Standards GB/T 18340.2–2010 [25], using

gas chromatography-mass spectrometry (GC-MS). GC was carried out in Portapak QCP-7551 columns at 50°C. The isotope ratio mass spectrometer was model MAT253, with an ion source voltage of 3.0 kV, current of 1.5 mA, and a precision of 0.4‰.

## 3. Results and discussions

### 3.1. Acid-hydrolyzed hydrocarbons and altered carbonates

Concentrations of methane (C1) and heavier hydrocarbons (C2+) in soil acid-hydrolyzed hydrocarbons (AHH) can be used to identify geochemical anomalies caused by hydrocarbons migration from oil and gas reservoirs [26]. In most soil samples collected **Acid-hydrolyzed hydrocarbons** in Xinchang gas field, the presence of n/i-butane and n/i-pentane can be detected. As shown in Figs 3(a)&(b), anomalies of methane and those of heavier hydrocarbons both present as sharp, distinctive spikes above the baseline and always coexist. The spikes within the range of the gas field (A11–18, B03–08, B11–13, and B20–27, indicated by underlined dotted bars in Fig 3) indicate that gaseous hydrocarbons are migrating from gas reservoirs and adsorbed by soil particles near surface. Meanwhile, the consecutive spikes outside the gas field (A40–A75, B38–41, B 44–50) suggest that oil and gas resources exist underneath these sites with anomalies detected. Detailed data for results reported in this section are available in the supporting information.

Part of hydrocarbons transported from reservoirs to surface can be oxidized by bacteria, resulting in the formation of altered carbonates (AC). Similar to AHH, formation of AC is considered a geochemical anomaly that can be used to delineate gas field area, predict reserve size, and trace hydrocarbon microseepage [20]. As shown in Figs 3(c)&(d), anomalies of AC concentration can be found along both Profile A and Profile B. Within the gas field range, AC concentration along Profile B forms several spiked patterns, consistent with those formed by AHH. However, the AC concentration along the A profile does not fluctuate greatly from the baseline. Outside the gas field, spikes of AC anomalies coexist with those of AHH. Overall, anomalies of AC and AHH are consistent, suggesting that hydrocarbons migrate from the gas reservoirs.

### 3.2. Surface methane Flux

**3.2.1. Methane flux measurement.** As illustrated in Fig 4, methane flux measurements in Xinchang gas field range from -0.123 to 0.022 mg/($m^2 \times d$), and the average flux is -0.009 mg/($m^2 \times d$). For the 23 test sites along Profile 1 (B2-35), 43% are measured with obvious negative flux of lower than -0.01 mg/($m^2 \times d$), and 57% with near-zero flux between -0.01 and 0.01 mg/($m^2 \times d$). For the 20 test sites along Profile 2 (C63-88), 20% are measured with positive flux of higher than 0.01 mg/($m^2 \times d$), 50% with obvious negative flux, and 30% with near-zero flux.

Comparison between measurements made along Profile 1 and Profile 2 finds little difference in average methane flux. However, all positive measurements occur along Profile 2. Note that test sites along Profile 2 are surrounded by a series of faults while those along Profile 1 are not (Fig 2), which confirms the theory that faults provide some favorable channels for methane migration from reservoirs to a few leaking points on the surface [27,28]. This finding is in accordance with previous studies [10,29].

Overall, while positive methane flux can be observed at a small fraction of test sites, the gas field in general is a sink of atmospheric methane. Note that the average background methane flux outside the gas field is -0.012 mg/($m^2 \times d$), which is lower than the average methane flux in the gas field. Therefore, the net methane flux from gas reservoirs to the surface in the area is estimated to be 0.003 mg/($m^2 \times d$), which partially offsets negative methane flux absorbed from the atmosphere.

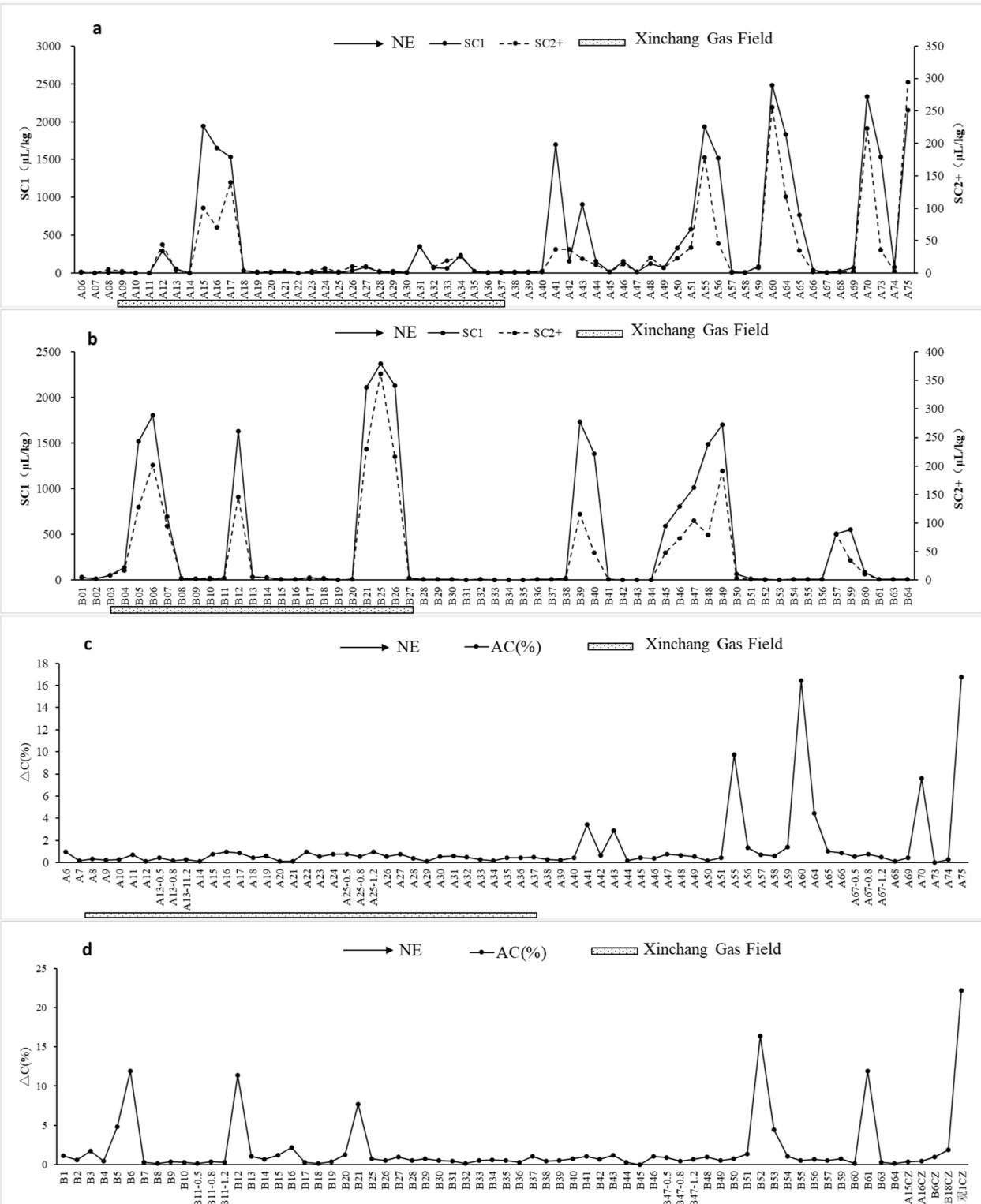

**Fig 3. Anomalies of methane, heavier hydrocarbons, and altered carbonates in Xinchang gas field.** (a) Methane (SC1) and heavier hydrocarbons (SC2+) along Profile A. (b) Methane (SC1) and heavier hydrocarbons (SC2+) along Profile B. (c) Altered carbonates (△C%) along Profile A. (d) Altered carbonates (△C%) along Profile B. Test sites from left to right in the figure extend towards northeast (NE) on the map.

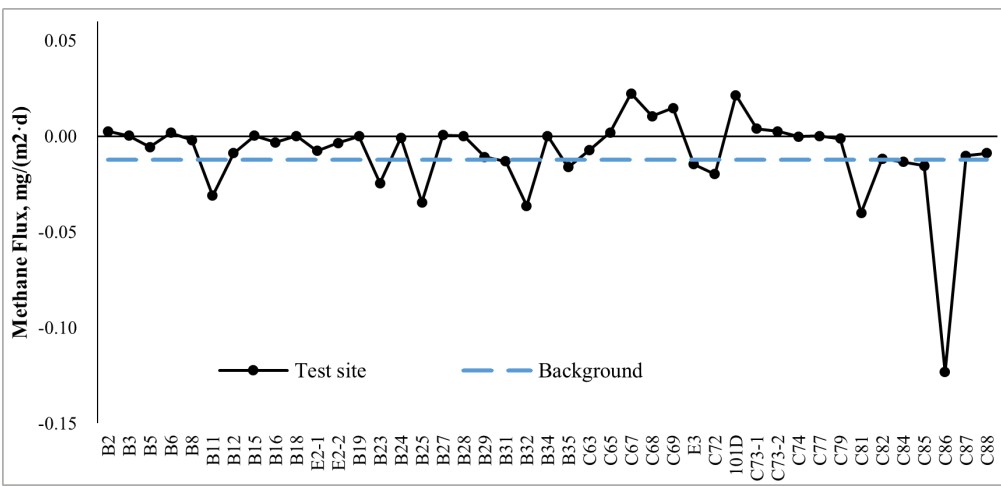

**Fig 4. Summary of methane flux measurements, in mg/(m$^2$ × d).**

**3.2.2. Methane flux patterns.** Based on the measurements made in Xinchang gas field, methane release from surface can be categorized as three kinds of patterns, namely (1) continuous, (2) episodic and (3) flat.

As shown in Fig 5(a), the continuous release pattern presents as continuous increment of methane concentration at a relatively stable rate, quickly reaching a higher concentration and stabilizing for a long period of time. This pattern is found to occur on test sites along Profile 2 in faulted zones. It is inferred that after methane is sealed by caprocks, the pressure accumulates and eventually allows for breakthrough; then along favorable channels in faults, methane emerges to surface against low resistance, manifesting as the pattern of continuous release.

As shown in Fig 5(b), the episodic release is characterized as spikes of methane concentration with inconstant strength at random intervals. Similar to the continuous release, the episodic release is also observed in faulted zones. However, the cause for the episodic release may differ from that for the continuous release in the sealing of caprocks – when methane is poorly sealed, it can migrate to the surface quickly on random disruption, resulting in sudden rise of methane flux, while lack of pressure accumulation leads to quick depletion of pulsed surges.

Lastly, as illustrated in Fig 5(c), the flat release pattern is characterized as flatlined flux, which indicates mass transfer of methane at constant rate, caused by stable methane concentration gradient and soil conditions such as porosity and moisture content.

### 3.3. Methane concentration and isotopic profiles in headspace and soil gas samples

Average and range of hydrocarbon concentrations in headspace gas samples are summarized in Table 1. The content of alkanes with two carbon atoms and more (C2+) measured in the background zone is very low, and the concentration of each alkane measured in the gas field is substantially higher than the background value. As C2+ alkanes are not common products of microbial activities, greater C2+ values suggest higher likelihood of hydrocarbon migration from reservoir to surface, i.e., microseepage.

As shown in Fig 6(a), methane concentrations in headspace gas samples in Xinchang gas field range from 4 to 29 ppm. Most concentrations measured in the test sites are higher than the background at the same depth. Whilst methane concentrations in most test sites do

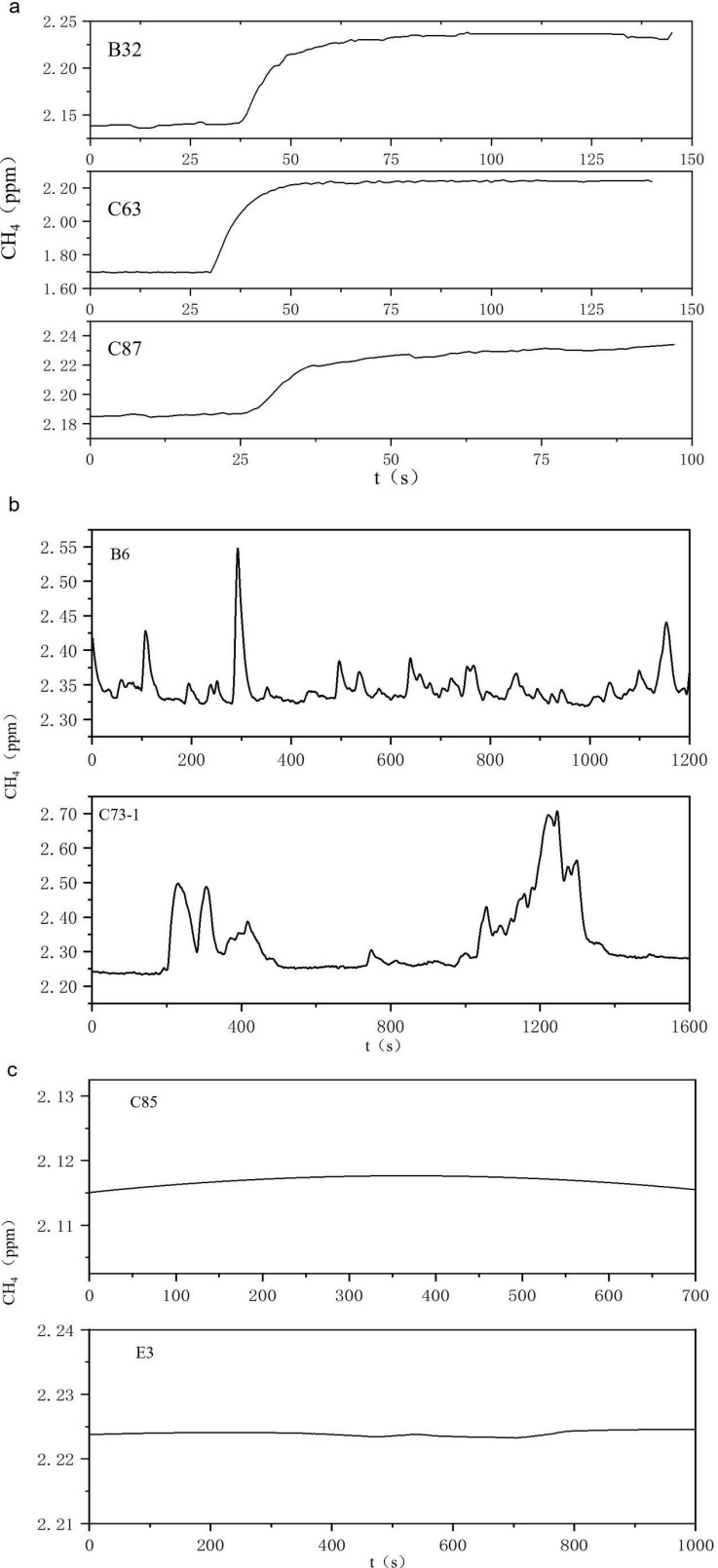

**Fig 5. Methane release patterns observed in Xinchang gas field.**

Table 1. Headspace gas composition in Xinchang gas field and background zone (ppm).

| Composition | Average, in gas field | Range, in gas field | Background |
|---|---|---|---|
| $CH_4$ | 21 | 4.0–29 | 3.5 |
| $C_2H_6$ | 0.59 | 0.05–6.8 | 0.08 |
| $C_3H_8$ | 0.24 | 0.02–2.3 | 0.04 |
| i-$C_4H_{10}$ | 0.05 | 0–0.44 | 0.03 |
| n-$C_4H_{10}$ | 0.06 | 0–0.61 | 0.02 |

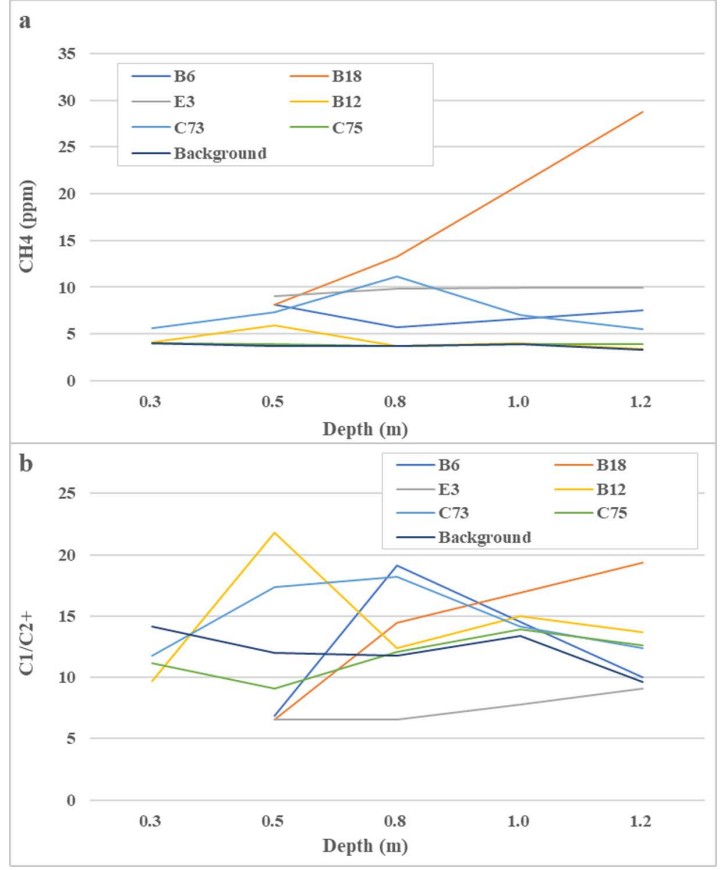

Fig 6. Methane concentration and C1/C2. + ratio in headspace samples versus depth (a.

not significantly vary as depth increases, a sharp rise is observed at Site B18. High methane concentration underground is a typical indication of strong methane migration from reservoir underneath. When methane moves towards to surface, it is gradually oxidized by microbial activities or atmosphere.

The methane coefficient (C1/C2+), defined as the ration between methane and C2+ alkanes, can help to reveal methane microseepage on the surface from oil and gas reservoirs underneath. Hydrocarbons with a methane coefficient of less than 50 are generally thermogenic from deeper sources [30]. As shown in Fig (b), methane coefficient values in headspace gas samples tested range from 6 to 22, suggesting thermogenic origin of the hydrocarbons detected.

Analysis results of the soil gas samples are summarized in Table 2. Methane concentrations in the soil gas samples retrieved in the gas field are at least one order of magnitude larger than those in the headspace gas samples, indicating strong methane adsorption by soil and rock particles, whilst methane coefficient values are all less than 50, which is in accordance with the results for headspace gas samples, confirming thermogenic origin of the hydrocarbons in the samples.

Carbon isotope ratios of methane ($\delta^{13}C_{CH4}$) range from $-42$ to $-21‰$ in soil gas samples, and $\delta^{13}C_{CO2}$ fall narrowly between -20 and -23‰. Typically, methane with $\delta^{13}C < -50‰$ is biogenic, namely formed by microbial activities, while that with $\delta^{13}C > -50‰$ is thermogenic; for carbon dioxide, $\delta^{13}C$ between -30 and -10‰ indicates biogenic origin [31,32]. Therefore, results of carbon isotopic analysis again confirm the thermogenic origin of methane in soil gas samples, which differs from the source of carbon dioxide.

### 3.4. Comparison with previous studies

Investigation of geological methane emissions have been carried out globally. A global dataset consisting of 1509 measurements sees methane microseepage flux ranging from 0.01 to 12 mg/($m^2 \times d$) in non-faulted areas, which is considered the lowest of the four activity levels; in comparison, methane seepage of the highest level can reach 300-1000 mg/($m^2 \times d$) [6]. In Indonesia, the Lusi hydrothermal system sees a methane seepage flux greater than 300 mg/($m^2 \times d$) [33]. In China, while methane microseepage from dry gas field has not been measured prior to this work, methane flux data for Dawanqi oil field [14] and Yakela condensate gas field [9] are available, which give average values of 17.4 and 2.9 mg/($m^2 \times d$), respectively. For methane seepage from coalbeds, emissions recorded from wellheads are typically very high [34], but in soil samples around coal mines, an average methane concentration of 10 ppm can be found, which is close to the results of this work [35].

Therefore, compared to methane flux data recorded globally, the net methane flux from underground to surface in Xinchang gas field (0.003 mg/($m^2 \times d$)) is considered extremely weak. According to the categorization of the global dataset mentioned above, the average methane flux in Xinchang fits into the lowest activity level, much lower methane flux than other geological features, such as the faulted or earthquake zone, oil seepage, gas-bearing spring, and mud volcanos, implying the methane reservoirs in Xinchang are well-sealed. Geologically, the slow methane migration in Xinchang is mainly due to the constraint factors of great depth of gas reservoirs and thick overburden of shale layers. In comparison, reservoirs in Dawanqi oil field are relatively shallow (<700 m) and permeable [14], so it is much easier for methane to escape. Similarly, reservoirs in Yakela condensate gas field are also poorly sealed, characterized by thin caprocks of low breakthrough pressure [36].

Nonetheless, despite the overall weaker flux, the three methane release patterns observed in Xinchang, as shown in Fig 5, mirror those observed in Dawanqi [37]. Given the differences

**Table 2. Soil gas composition and carbon isotopic ratio in Xinchang gas field and background zone.**

| Test sites | CH$_4$ (PPM) | C1/C2+ | $\delta^{13}C_{CH4}$ (‰) | $\delta^{13}C_{CO2}$ (‰) |
|---|---|---|---|---|
| B8 | 343 | 30.45 | -28 | -20 |
| B18 | 288 | 28.38 | -30 | -20 |
| B24 | 302 | 11.01 | -21 | -22 |
| B34 | 885 | 5.46 | -24 | -23 |
| C63 | 495 | 18.13 | -37 | -20 |
| E3 | 540 | 27.82 | -32 | -21 |
| Background | 282 | 191.99 | -42 | -21 |

in geological condition, environment, and reservoir type between the Xinchang and Dawanqi fields, the observation of these three patterns in both fields suggests that these phenomena may be universal. The manifestation of these patterns has shown close connection with factors influencing methane migration, such as fault structures, caprock sealing, and soil porosity, highlighting the need for further investigation to fully understand the complex dynamics.

Interestingly, Xinchang gas field is identified as a net sink of atmospheric methane, indicating notable rate of methane consumption in the soil, which could be attributed to climatic factors. As stated in the Introduction Section, Xinchang gas field is located in mild and humid climate, which supports a rich diversity of fauna and flora and facilitating active methanotrophic microbial activities. Therefore, the negative methane flux for the background data in Xinchang is in accordance with the general consideration that, in a warm and humid climatic environment, methanotrophic consumption becomes dominant and can significantly reduce methane flux [1,11]. Meanwhile, as Xinchang sees daily average temperature below 0 throughout the year, its winter is never so cold, as to form snow cover that may block methane from entering the atmosphere [10]. Therefore, it is suggested that Xinchang field will see higher methane flux in the winter than summer. On the other hand, both the Dawanqi and Yakela fields in Tarim Basin (cold and dry, desert), see much higher methane fluxes than Xinchang. In such extreme desert conditions, methane consumption by soil microbial activities is negligible. The discussion above implies the influence of local climate and biological factors on soil methane balance, calling for closer investigation on activities and genomic characteristics of microbial community in confirmed and potential hotspots of geological methane emissions in future research, which can substantially contribute to the understanding of overall methane flux from the soil and management of the atmospheric methane level. Meanwhile, in other areas with extreme climatic conditions, such as Alaska (cold and dry, polar), Venezuela (hot and humid, tropical), and Arabian Peninsula (hot and dry, desert), methane fluxes much higher than Xinchang gas field have been recorded [6]. Therefore, it is suggested that the climatic condition does not play a determining role in magnitude of methane microseepage, and that the investigation of microseepage behavior calls for thorough and comprehensive analysis of geological, climatic, and microbial factors.

Climate change may also in turn impact methane microseepage flux. Should global temperature moderately rise, the Xinchang area, as well as many other oil and gas reservoirs located in temperate, humid climate, would see a change in favor of microbial activity, potentially enhancing the role of soil as a sink of methane. On the other hand, if global temperature rises to dramatically, combined with the fact that global climate is a chaotic system [38], local climate may be drastically changed, potentially to a hot, arid one that would eliminate the majority of microbial methane consumptions. Therefore, in future research, it is important to investigate how future climate scenarios might alter microbial activities and affect methane microseepage.

## 3.5. Policy implications and potential improvements

Manifestations of methane microseepage from natural gas reservoirs leads to several strong policy implications on the intervention and management of geological methane emissions. First and foremost, it is crucial to enhance the monitoring of methane emissions from oil and gas facilities, not only from mining, processing, and transporting, but also from peripheral areas in the form of microseepage. Researchers have estimated that fugitive emissions from the natural gas industry amount to 2–4% of the total gas produced [39], calling for stringent regulation on the emissions from the industry. This urgency will only be underscored by the confirmation of methane microseepage from the reservoir areas.

Additionally, it is to be noted that methane microseepage can be discontinuous both spatially and temporally, as complicated, unpredictable geological microstructures can have complex impact on methane migration behaviors. Meanwhile, short-term random changes and periodic variations in weather conditions, such as local temperature and humidity, can also affect soil microbial activity levels and ultimately impact methane absorption by soil. For instance, methane flux measured in relatively warmer and wetter seasons can be lower than in colder and drier seasons, as the former conditions can facilitate microbial methanotrophic oxidation in soil [11]. These factors increase the degree of uncertainty in short-term in-situ methane flux monitoring. Consequently, to mitigate such spatial and temporal uncertainty and bias introduced by short-term in-situ monitoring, long-term effort over vast reservoir areas would be needed. This would also provide insights to better understanding long-term variation patterns of methane microseepage, on how it may periodically change or trend over a longer timescale.

However, note that long-term in-situ monitoring would be both expensive and labor-intensive. It is thus implied that remote sensing [40] should be coupled with in-situ methods to improve the accuracy of methane microseepage surveillance and provide a more comprehensive picture of microseepage flux. Remote sensing technologies can remarkably facilitate the surveillance process by providing a timelier and more holistic overview of atmospheric methane concentration distribution on national or regional scale, pinpointing where atmospheric methane anomaly occurs. However, remote sensing technologies still present a great degree of uncertainties, as there exist difficulties to determine specific methane sources, as well as large gaps between bottom-up estimation and top-down measurements [41]. Therefore, in-situ and remote sensing measurements may be combined to increase accuracy. With regional methane concentration anomaly identified, in-situ monitoring can now serve as the means of sampling and calibration to identify exact methane microseepage hotspots and thus further increase surveillance effectiveness.

Lastly, it is revealed that methane microseepage is subject to the impact by climate conditions and ultimately microbial activities. While this impact is not the central focal point of this work, it nonetheless indicates that development of methane emission intervention and management policies should consider such factors and calls for closer scrutiny and analysis in the future.

## 4. Conclusion

In this study, surface methane flux in Xinchang gas field is monitored. The gas field is overall a weak sink of atmospheric methane, but low level of methane microseepage from natural gas reservoirs can be confirmed. Methane release patterns can be categorized as (1) continuous, (2) episodic, and (3) flat, which is accordance with observation made in other oil and gas fields. Soil samples are collected for analysis of acid-hydrolyzed hydrocarbon composition and altered carbonate concentration. Spikes of geochemical anomalies in these indicators co-exist both within and outside the range of the gas field, suggesting hydrocarbon migration from reservoirs to earth surface. This finding is further confirmed by analyzing hydrocarbon concentrations and carbon isotopic profiles of headspace and soil gas samples, which all indicate that methane and heavier hydrocarbons in the samples are of thermogenic origin, namely from reservoirs underneath. Overall, the case study of Xinchang gas field provides new evidence for the behavior of methane microseepage from natural gas reservoirs in mild climate. The results confirm that magnitude and pattern of methane microseepage are dictated by such factors as fault structure, caprock sealing, and reservoir depth, and that climate also plays a critical role in methane absorption from atmosphere to soil. Based on these findings, it is also

suggested that close monitoring and stringent regulation of methane microseepage, as well as continuous investigation on factors affecting this phenomenon, are essential to the management of geological methane emissions. Therefore, this paper offers valuable insights applicable to methane microseepage management in oil and gas reservoirs in other regions.

## Author contributions

**Data curation:** Zhourong Ke, Weiwei Ji, Caixin Pu.

**Formal analysis:** Zhourong Ke, Guojian Wang.

**Funding acquisition:** Huimin Yun, Junhong Tang.

**Investigation:** Maoyi Tang, Guojian Wang.

**Methodology:** Junhong Tang.

**Resources:** Huimin Yun, Guojian Wang.

**Software:** Maoyi Tang, Zhourong Ke, Chongxin Huang.

**Supervision:** Haoqi Wang, Huimin Yun, Junhong Tang.

**Validation:** Chongxin Huang.

**Visualization:** Maoyi Tang, Weiwei Ji.

**Writing – original draft:** Zhourong Ke.

**Writing – review & editing:** Haoqi Wang, Huimin Yun.

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
