## [Decision Letter · Decision Letter 0]

12 Mar 2024

PONE-D-24-03564Characterization of methane microseepage from natural gas reservoirs in mild climate: a case study of Xinchang gas fieldPLOS ONE

Dear Dr. Wang,

Thank you for submitting your manuscript to PLOS ONE. After careful consideration, we feel that it has merit but does not fully meet PLOS ONE’s publication criteria as it currently stands. Therefore, we invite you to submit a revised version of the manuscript that addresses the points raised during the review process.

We look forward to receiving your revised manuscript.

Kind regards,

Trung Quang Nguyen

Academic Editor

PLOS ONE

Journal Requirements:

   "Natural Science Foundation of China (NSFC Grants No. U2003101, 41872126, and 41373121).

Hangzhou Dianzi University (Grants No. KYS205622065)."

5. In the online submission form, you indicated that "Data underlying the results presented in the study are available by request for researchers who meet the criteria for access to confidential data."

7. We note that Figure 1 in your submission contain map/satellite images which may be copyrighted. All PLOS content is published under the Creative Commons Attribution License (CC BY 4.0), which means that the manuscript, images, and Supporting Information files will be freely available online, and any third party is permitted to access, download, copy, distribute, and use these materials in any way, even commercially, with proper attribution. For these reasons, we cannot publish previously copyrighted maps or satellite images created using proprietary data, such as Google software (Google Maps, Street View, and Earth). For more information, see our copyright guidelines: http://journals.plos.org/plosone/s/licenses-and-copyright.

Reviewers' comments:

Reviewer's Responses to Questions

**Comments to the Author**

1. Is the manuscript technically sound, and do the data support the conclusions?

Reviewer #1: Yes

Reviewer #2: Yes

2. Has the statistical analysis been performed appropriately and rigorously? 

Reviewer #1: Yes

Reviewer #2: Yes

3. Have the authors made all data underlying the findings in their manuscript fully available?

Reviewer #1: Yes

Reviewer #2: Yes

4. Is the manuscript presented in an intelligible fashion and written in standard English?

Reviewer #1: Yes

Reviewer #2: Yes

5. Review Comments to the Author

Reviewer #1: This document provides additional comments and considerations for the manuscript titled "Characterization of methane microseepage from natural gas reservoirs in mild climate: a case study of Xinchang gas field." The comments focus on potential concerns regarding duplicate publication, research ethics, and publication ethics.

Duplicate Publication: There is no indication that the results presented have been published elsewhere, which is positive. However, the authors should explicitly clarify if any part of the work, including data, figures, or substantial portions of the text, has been or is under consideration for publication in another venue. This clarification is essential to avoid issues of self-plagiarism and duplicate publication.

Research Ethics: The methodology section provides detailed information on data collection and analysis. Nonetheless, it would enhance the manuscript's integrity if the authors included a statement on ethical considerations, especially regarding environmental impact assessments of the study area. Given the environmental sensitivity of methane emissions, outlining measures taken to minimize potential harm or disturbance during the study would be beneficial.

Publication Ethics: The manuscript should adhere to high standards of publication ethics, including full transparency about the contributions of each author and the disclosure of any potential conflicts of interest. While the manuscript includes funding information, a more detailed statement regarding the authors' adherence to ethical guidelines in conducting and reporting their research is recommended.

In conclusion, while the study presents valuable insights into methane microseepage from natural gas reservoirs in a mild climate, addressing the above concerns can significantly strengthen the manuscript. Ensuring clarity on these aspects would enhance the manuscript's contribution to the field and its ethical and scientific rigor.

The study on "Characterization of methane microseepage from natural gas reservoirs in mild climate: a case study of Xinchang gas field" provides significant insights into methane emissions and their implications for climate change. The comprehensive analysis of methane flux, hydrocarbon composition, and carbon isotope profiles in the Xinchang gas field is commendable for its depth and detail. Here are some additional comments and suggestions for further strengthening your manuscript:

Integration with Global Data: While your study provides critical data on methane microseepage in the Xinchang gas field, integrating these findings with global datasets on methane emissions could offer a broader perspective on the significance of your results. Comparing your data with methane flux measurements from different geological settings worldwide could contextualize the unique aspects of methane microseepage in mild climates.

Methodological Advancements: Your methodology, combining field measurements and laboratory analysis, is robust. However, discussing any potential limitations of your techniques and how they compare with other contemporary methods could enhance the reliability of your findings. For instance, detailing the calibration process of the greenhouse gas analyzer and static gas collection chambers could provide clarity on the accuracy of your measurements.

Microbial Activity Analysis: Given the significant role of microbial activities in methane consumption, a more detailed analysis of the microbial communities present in the soil samples could provide insights into the biodegradation processes affecting methane flux. This could involve genomic or metagenomic analyses to identify methanotrophic communities and their functional capabilities.

Impact of Climate Change: Considering the implications of methane on climate change, an analysis of how future climate scenarios might affect methane microseepage patterns in the Xinchang gas field and similar environments would be valuable. This could involve modeling studies to predict changes in methane flux under various climate change projections.

Policy and Management Implications: Expanding the discussion on the implications of your findings for methane management and climate change mitigation policies could make your study more impactful. Suggestions for policy interventions or management strategies to mitigate methane emissions from natural gas reservoirs could provide practical value to your research.

Longitudinal Studies: Mentioning the potential for longitudinal studies to observe changes in methane microseepage over time, especially in the context of ongoing climate change and human interventions in natural gas fields, could highlight the importance of continued research in this area.

These comments are intended to complement the strengths of your manuscript and suggest avenues for further enhancement. Your research contributes significantly to our understanding of methane emissions from natural gas reservoirs and their climate implications, highlighting the need for comprehensive monitoring and management strategies.

Reviewer #2: Please double check the manuscript and revise it carefull before re-submit to the Journal:

- The Section 2. Geological 76 Settings of Xinchang Gas Field should be included in the Section 1. Introduction.

- The Section 3.2. Laboratory Analysis should be cited some references for the methods used

- Please update the reference to papers published in the last 5 years.

- There are several DOI does not work, please check for any error in the links.

- Some references without DOI, which should be added; and some references were listed with Capital character: Ex. Ref 17, 19…

- The conclusion section should not separate into different paragraphs as in the present manuscript version.

- This phrase should not use in the conclusion section: “The results are consistent with previous studies…”.

6. PLOS authors have the option to publish the peer review history of their article (what does this mean? ). If published, this will include your full peer review and any attached files.

**Do you want your identity to be public for this peer review?** For information about this choice, including consent withdrawal, please see our Privacy Policy .

Reviewer #1: No

Reviewer #2: No

---

## [Author Response · Author response to Decision Letter 1]

22 Jul 2024

Response: We have edited our manuscript based on the PLOS ONE requirements.

Response: No permits were required. The field work was conducted on public lands surrounding the gas wells.

Response: Removed.

"Natural Science Foundation of China (NSFC Grants No. U2003101, 41872126, and 41373121).

Hangzhou Dianzi University (Grants No. KYS205622065)."

Response: We have included the amended Role of Funder statement in our cover letter

5. In the online submission form, you indicated that "Data underlying the results presented in the study are available by request for researchers who meet the criteria for access to confidential data."

Response: We have uploaded the data in supplementary information.

Response: We have uploaded the data in supplementary information, available to all readers.

7. We note that Figure 1 in your submission contain map/satellite images which may be copyrighted. All PLOS content is published under the Creative Commons Attribution License (CC BY 4.0), which means that the manuscript, images, and Supporting Information files will be freely available online, and any third party is permitted to access, download, copy, distribute, and use these materials in any way, even commercially, with proper attribution. For these reasons, we cannot publish previously copyrighted maps or satellite images created using proprietary data, such as Google software (Google Maps, Street View, and Earth). For more information, see our copyright guidelines: http://journals.plos.org/plosone/s/licenses-and-copyright.

Response: We have redrawn the figure to avoid any potential copyright issue.

Reviewers’ Comments

Reviewer #1: This document provides additional comments and considerations for the manuscript titled "Characterization of methane microseepage from natural gas reservoirs in mild climate: a case study of Xinchang gas field." The comments focus on potential concerns regarding duplicate publication, research ethics, and publication ethics.

Duplicate Publication: There is no indication that the results presented have been published elsewhere, which is positive. However, the authors should explicitly clarify if any part of the work, including data, figures, or substantial portions of the text, has been or is under consideration for publication in another venue. This clarification is essential to avoid issues of self-plagiarism and duplicate publication.

Response: We hereby clarify that no part of this work, including its data, figures, and texts in length, has been published or under consideration for publication in another venue.

Research Ethics: The methodology section provides detailed information on data collection and analysis. Nonetheless, it would enhance the manuscript's integrity if the authors included a statement on ethical considerations, especially regarding environmental impact assessments of the study area. Given the environmental sensitivity of methane emissions, outlining measures taken to minimize potential harm or disturbance during the study would be beneficial.

Response: This field work involved in this manuscript include methane flux monitoring and sampling in shallow earth (1.5m). These activities do not cause additional methane emissions. Disturbance on local flora, fauna, habitat, and water bodies are minimal. Therefore, we are confident to state that environmental impacts on the study area are negligible.

To our knowledge, it doesn’t seem customary for such statement to appear in the manuscript. If it needs to be, please let us know.

Publication Ethics: The manuscript should adhere to high standards of publication ethics, including full transparency about the contributions of each author and the disclosure of any potential conflicts of interest. While the manuscript includes funding information, a more detailed statement regarding the authors' adherence to ethical guidelines in conducting and reporting their research is recommended.

In conclusion, while the study presents valuable insights into methane microseepage from natural gas reservoirs in a mild climate, addressing the above concerns can significantly strengthen the manuscript. Ensuring clarity on these aspects would enhance the manuscript's contribution to the field and its ethical and scientific rigor.

Response: Our statement to dual submission and plagiarism has been provided in the first response. This work does not involve any human or animal research, so there is nothing to declare on this aspect. We herby declare, again, that this work adheres to all ethical guidelines of PLOS.

And again, to our knowledge, it doesn’t seem customary for such statement to appear in the manuscript. If it needs to be, please let us know.

The study on "Characterization of methane microseepage from natural gas reservoirs in mild climate: a case study of Xinchang gas field" provides significant insights into methane emissions and their implications for climate change. The comprehensive analysis of methane flux, hydrocarbon composition, and carbon isotope profiles in the Xinchang gas field is commendable for its depth and detail. Here are some additional comments and suggestions for further strengthening your manuscript:

Integration with Global Data: While your study provides critical data on methane microseepage in the Xinchang gas field, integrating these findings with global datasets on methane emissions could offer a broader perspective on the significance of your results. Comparing your data with methane flux measurements from different geological settings worldwide could contextualize the unique aspects of methane microseepage in mild climates.

Response: We have elaborated on the comparison of this study with global dataset in Section 3.4.

Methodological Advancements: Your methodology, combining field measurements and laboratory analysis, is robust. However, discussing any potential limitations of your techniques and how they compare with other contemporary methods could enhance the reliability of your findings. For instance, detailing the calibration process of the greenhouse gas analyzer and static gas collection chambers could provide clarity on the accuracy of your measurements.

Response: The calibration of the GHG analyzer is against the background concentrations in air.

Microbial Activity Analysis: Given the significant role of microbial activities in methane consumption, a more detailed analysis of the microbial communities present in the soil samples could provide insights into the biodegradation processes affecting methane flux. This could involve genomic or metagenomic analyses to identify methanotrophic communities and their functional capabilities.

Response: We agree with this comment that microbial activity is critical to methane consumption. Our research has found out confirm negative background methane flux, suggesting in warm climate where microbial communities are active, the soil acts as a sink of methane. Genomic or metagenomic analyses will certainly provide insights in the functionality of the microbial community in Xinchang gas area, but this is already beyond the scope of our project and expertise in the field of chemistry and environmental science. To the purpose of this project, identifying the ensemble effect of the microbial community on methane flux should suffice. We will leave the follow-up to future research and apply for new fundings for it to happen.

Impact of Climate Change: Considering the implications of methane on climate change, an analysis of how future climate scenarios might affect methane microseepage patterns in the Xinchang gas field and similar environments would be valuable. This could involve modeling studies to predict changes in methane flux under various climate change projections.

Response: We certainly agree that climate change and GHG sources can have mutual impact on each other. One example is warm weather and mountain pine beetle infestation – warm weather increases the survival rate of the beetle community in winter, which in turn cause more severe destruction of pines and wildfires out of dead, dry trees.

We added some discussion on the potential impact of climate on methane microseepage, most notably on microbial activity and its role as methane sink, in the end of Section 3.4. However, analysis on a detailed fashion is beyond our expertise, and its scope is big enough for many more publications.

Policy and Management Implications: Expanding the discussion on the implications of your findings for methane management and climate change mitigation policies could make your study more impactful. Suggestions for policy interventions or management strategies to mitigate methane emissions from natural gas reservoirs could provide practical value to your research.

Response: Policy implications have been added to the new Section 3.5.

Longitudinal Studies: Mentioning the potential for longitudinal studies to observe changes in methane microseepage over time, especially in the context of ongoing climate change and human interventions in natural gas fields, could highlight the importance of continued research in this area.

Response: We agree with this comment and are more than happy to follow up on observation over time. To our knowledge, seasonable variation clearly exists for methane microseepage. Monitoring methane emissions over a long period of time are also critical to understanding the evolving climate forcing and continuous methane management.

However, as the project that supports this work has concluded, we will have to apply for new fundings for the follow-up to happen.

These comments are intended to complement the strengths of your manuscript and suggest avenues for further enhancement. Your research contributes significantly to our understanding of methane emissions from natural gas reservoirs and their climate implications, highlighting the need for comprehensive monitoring and management strategies.

Response: Thank you very much.

Reviewer #2: Please double check the manuscript and revise it carefull before re-submit to the Journal:

- The Section 2. Geological 76 Settings of Xinchang Gas Field should be included in the Section 1. Introduction.

Response: Section 2 has merged into Section 1.

- The Section 3.2. Laboratory Analysis should be cited some references for the methods used

Response: Methods used for flux monitoring and lab analysis have been properly cited.

- Please update the reference to papers published in the last 5 years.

Response: In the field of methane microseepage, there aren’t that many relevant papers in the most recent years. We have added as many relevant ones as w

---

## [Decision Letter · Decision Letter 1]

14 Aug 2024

PONE-D-24-03564R1Characterization of methane microseepage from natural gas reservoirs in mild climate: a case study of Xinchang gas fieldPLOS ONE

Dear Dr. Wang,

Thank you for submitting your manuscript to PLOS ONE. After careful consideration, we feel that it has merit but does not fully meet PLOS ONE’s publication criteria as it currently stands. Therefore, we invite you to submit a revised version of the manuscript that addresses the points raised during the review process.

We look forward to receiving your revised manuscript.

Kind regards,

Trung Quang Nguyen

Academic Editor

PLOS ONE

Journal Requirements:

Reviewers' comments:

Reviewer's Responses to Questions

**Comments to the Author**

1. If the authors have adequately addressed your comments raised in a previous round of review and you feel that this manuscript is now acceptable for publication, you may indicate that here to bypass the “Comments to the Author” section, enter your conflict of interest statement in the “Confidential to Editor” section, and submit your "Accept" recommendation.

Reviewer #1: All comments have been addressed

Reviewer #3: All comments have been addressed

2. Is the manuscript technically sound, and do the data support the conclusions?

Reviewer #1: Yes

Reviewer #3: Yes

3. Has the statistical analysis been performed appropriately and rigorously? 

Reviewer #1: Yes

Reviewer #3: Yes

4. Have the authors made all data underlying the findings in their manuscript fully available?

Reviewer #1: Yes

Reviewer #3: Yes

5. Is the manuscript presented in an intelligible fashion and written in standard English?

Reviewer #1: Yes

Reviewer #3: Yes

6. Review Comments to the Author

Reviewer #1: The author's response to the feedback provided in the initial review demonstrates a commendable effort to address the reviewers' concerns and significantly improve the manuscript. The revisions made, particularly in the methodological section and the integration of additional references, have enhanced the scientific rigor of the study. The addition of a detailed comparison with global datasets on methane emissions strengthens the contextualization of the study's findings, allowing for a broader application of the results. Furthermore, the clarification regarding ethical considerations and environmental impacts contributes to the transparency and credibility of the research. The response to the suggestions on the microbial activity analysis and the potential impacts of climate change reflects a thoughtful approach to acknowledging the study's limitations while outlining future research directions. The restructuring of certain sections and the refinement of the conclusion have improved the manuscript's overall coherence and clarity. In summary, the authors have successfully addressed the critical points raised, resulting in a more robust and scientifically sound manuscript that makes a valuable contribution to the field.

Reviewer #3: I would advise authors to mention some recommendations at the end of both abstract and conclusion based on the results of this study regarding current and future practices in the subjects of this study.

7. PLOS authors have the option to publish the peer review history of their article (what does this mean? ). If published, this will include your full peer review and any attached files.

**Do you want your identity to be public for this peer review?** For information about this choice, including consent withdrawal, please see our Privacy Policy .

Reviewer #1: No

Reviewer #3: No

---

## [Author Response · Author response to Decision Letter 2]

13 Sep 2024

Reviewer #1: The author's response to the feedback provided in the initial review demonstrates a commendable effort to address the reviewers' concerns and significantly improve the manuscript. The revisions made, particularly in the methodological section and the integration of additional references, have enhanced the scientific rigor of the study. The addition of a detailed comparison with global datasets on methane emissions strengthens the contextualization of the study's findings, allowing for a broader application of the results. Furthermore, the clarification regarding ethical considerations and environmental impacts contributes to the transparency and credibility of the research. The response to the suggestions on the microbial activity analysis and the potential impacts of climate change reflects a thoughtful approach to acknowledging the study's limitations while outlining future research directions. The restructuring of certain sections and the refinement of the conclusion have improved the manuscript's overall coherence and clarity. In summary, the authors have successfully addressed the critical points raised, resulting in a more robust and scientifically sound manuscript that makes a valuable contribution to the field.

Response: Thanks very much for the comment.

Reviewer #3: I would advise authors to mention some recommendations at the end of both abstract and conclusion based on the results of this study regarding current and future practices in the subjects of this study.

Response: A few recommendations have been added to both abstract and conclusion.

---

## [Decision Letter · Decision Letter 2]

18 Sep 2024

PONE-D-24-03564R2Characterization of methane microseepage from natural gas reservoirs in mild climate: a case study of Xinchang gas fieldPLOS ONE

Dear Dr. Wang,

Thank you for submitting your manuscript to PLOS ONE. After careful consideration, we feel that it has merit but does not fully meet PLOS ONE’s publication criteria as it currently stands. Therefore, we invite you to submit a revised version of the manuscript that addresses the points raised during the review process.

We look forward to receiving your revised manuscript.

Kind regards,

Trung Quang Nguyen

Academic Editor

PLOS ONE

Reviewers' comments:

Reviewer's Responses to Questions

**Comments to the Author**

1. If the authors have adequately addressed your comments raised in a previous round of review and you feel that this manuscript is now acceptable for publication, you may indicate that here to bypass the “Comments to the Author” section, enter your conflict of interest statement in the “Confidential to Editor” section, and submit your "Accept" recommendation.

Reviewer #1: All comments have been addressed

2. Is the manuscript technically sound, and do the data support the conclusions?

Reviewer #1: Yes

3. Has the statistical analysis been performed appropriately and rigorously? 

Reviewer #1: Yes

4. Have the authors made all data underlying the findings in their manuscript fully available?

Reviewer #1: Yes

5. Is the manuscript presented in an intelligible fashion and written in standard English?

Reviewer #1: Yes

6. Review Comments to the Author

Reviewer #1: The study on methane microseepage from the Xinchang gas field provides valuable insights into geological methane emissions in mild climates. The methodology, including soil sampling and methane flux monitoring, is thorough and appropriate for identifying geochemical anomalies and migration patterns. However, the paper could improve its discussion on the limitations of the study, particularly regarding the impact of local climate variability on methane flux. The absence of long-term monitoring limits the understanding of temporal variations in microseepage. Additionally, the discussion could further explore the role of microbial activity in methane oxidation, which may significantly influence surface flux measurements. Furthermore, while the use of static gas collection chambers is common, complementing these measurements with remote sensing data could enhance spatial coverage and detection sensitivity. The study's findings align with existing literature, yet it would be beneficial to compare the results with regions having different climatic conditions to provide a more comprehensive understanding of methane microseepage behavior.

Questions:

1. How might seasonal variations in temperature and precipitation affect methane microseepage in the Xinchang gas field?

2. What are the potential biases introduced by short-term methane flux measurements, and how can they be mitigated?

3. How could the integration of remote sensing techniques improve the detection of methane microseepage compared to traditional methods?

4. In what ways does microbial activity in soil influence the measurement of methane flux, and how was this accounted for in the study?

5. How do the results from the Xinchang gas field compare to other regions with different geological or climatic conditions?

7. PLOS authors have the option to publish the peer review history of their article (what does this mean? ). If published, this will include your full peer review and any attached files.

**Do you want your identity to be public for this peer review?** For information about this choice, including consent withdrawal, please see our Privacy Policy .

Reviewer #1: No

---

## [Author Response · Author response to Decision Letter 3]

14 Oct 2024

Reviewer #1: The study on methane microseepage from the Xinchang gas field provides valuable insights into geological methane emissions in mild climates. The methodology, including soil sampling and methane flux monitoring, is thorough and appropriate for identifying geochemical anomalies and migration patterns. However, the paper could improve its discussion on the limitations of the study, particularly regarding the impact of local climate variability on methane flux. The absence of long-term monitoring limits the understanding of temporal variations in microseepage. Additionally, the discussion could further explore the role of microbial activity in methane oxidation, which may significantly influence surface flux measurements. Furthermore, while the use of static gas collection chambers is common, complementing these measurements with remote sensing data could enhance spatial coverage and detection sensitivity. The study's findings align with existing literature, yet it would be beneficial to compare the results with regions having different climatic conditions to provide a more comprehensive understanding of methane microseepage behavior.

Response:

Thanks for the constructive comments.

The authors completely agree that long-term monitoring could provide more insights on the evolvement of methane microseepage over time. However, this project has already been ended, so the follow-up monitoring could only be conducted once new fundings can be secured. Meanwhile, there are other studies that focused on seasonal variations in methane microseepage, forming reasonable theories that are also applicable to this case. Therefore, the authors have now developed more discussions on potential seasonal variations in methane microseepage in Xinchang.

Regarding microbial activity in methane oxidation, it certainly has significant impact on methane flux. In this work, the authors have taken it into account, by measuring background methane flux outside the test zones. This background flux is mainly due to the ensemble effect of microbe living in soil. Meanwhile, the authors have also conducted tests to rule out the possibility of notable biogenic methane in the soil. In this way, the authors feel it is safe to draw a high-level conclusion on the role of microbial activity on methane flux, and on the existence and extent of thermogenic methane microseepage from the gas reservoir. Naturally, developing an understanding on the exact methane-related behavior of soil microorganisms are crucial to methane management, yet it is outside the scope of this work and beyond the engineering expertise of the authors. The authors have pointed this topic out, as potential future work for interested microbiologists.

Lastly, the authors agree that complementing in-situ monitoring with remote sensing could enhance the effectiveness of methane monitoring. Yet again, remote sensing would be a whole new project to this one, so the authors provide some brief discussion on it. The authors have also provide comparison of this work with previous ones on different geological and climatic conditions.

Overall, the authors completely agree with the constructively comments provided by the reviewer. Addressing these limitations would certainly improve the general effectiveness in methane flux monitoring, but the work will be beyond the scope of this work and require multidisciplinary expertise. Therefore, the authors have made every effort possible to provide detailed discussions on these limitations of this study and shed light on potential future work.

Questions:

1. How might seasonal variations in temperature and precipitation affect methane microseepage in the Xinchang gas field?

Response: As suggested by a series of studies, such as ref [10] and [11] cited in this manuscript, temperature and precipitation may affect methane microseepage flux in two competing ways. On one hand, cold temperature and arid weather tend to hamper microbial methanotrophic consumptions, which increases net methane flux, as observed in [11]. On the other hand, extreme cold weather may lead to snow covers thick enough to block methane flux, as observed in [10]. These two factors are competing with each other, leading to dynamic seasonal variation patterns in different regions.

As stated in Section 1, Xinchang field is located in a mild climate that never sees daily average temperature below 0 throughout the year. It has a relatively cold and dry winter, which leads to lower microbial activities and thus higher methane flux, yet is never cold enough to allow for the presence of snow cover that blocks methane from entering the atmosphere. Therefore, it is suggested that Xinchang field will see higher methane flux in the winter.

2. What are the potential biases introduced by short-term methane flux measurements, and how can they be mitigated?

Response: As revealed in this work and many others, short-term methane flux can manifest in various patterns. This can be explained by complicated geological micro-structures. As a result, short-term methane flux may come with a great extent of uncertainty, which may be mitigated in two ways.

Firstly, long-term monitoring over a large area may be executed. This would lead to more accurate and thorough monitoring, but is naturally expensive and labor-intensive. Secondly, in-situ monitoring may be coupled with remote sensing techniques, which leads to the third comment point of the reviewer.

Thus, the authors have expanded the section 3.5 in the manuscript regarding this and the next comment of the reviewer.

3. How could the integration of remote sensing techniques improve the detection of methane microseepage compared to traditional methods?

Response: Remote sensing can be coupled with traditional in-situ monitoring to improve the accuracy of methane microseepage surveillance. Traditional in-situ monitoring over a large region can be labor-intensive and expensive. In this sense, remote sensing can effectively provide an overview of atmospheric methane concentration on national or regional scale, pinpointing where atmospheric methane anomaly occurs. With the guidance of remote sensing results, in-situ monitoring can now be used to identify exact methane microseepage hotspots in an area with known abnormal atmospheric methane concentration distribution.

4. In what ways does microbial activity in soil influence the measurement of methane flux, and how was this accounted for in the study?

Response: As stated in Section 3.3, both methane coefficient (C1/C2+) and Carbon isotope ratio, analyzed in this work, can help to identify the origin of methane and other hydrocarbons. The results of both indicators suggest that the methane and hydrocarbons sampled from our test sites are thermogenic, instead of biogenic. That is to say, this hydrocarbon mix, methane included, originates from underground fossil fuel reservoir, rather than from living microorganisms in soil.

On the other hand, the background methane flux recorded outside the gas field zone in our work is negative. This is clearly a sign of methane absorption from atmosphere to soil, by no other means than microbial activities.

Naturally, to determine the methane microseepage flux, that is the methane migrated from reservoir to atmosphere, the background methane flux is deducted from the flux measured on test sites.

5. How do the results from the Xinchang gas field compare to other regions with different geological or climatic conditions?

Response: The entire Section 3.4 discusses comparison of this study with previous ones. We have reviewed ten studies on this topic and compared the results with Xinchang field. In short, the net methane flux from Xinchang gas field is considered very weak, compared to other oil and gas reservoirs. The mild and humid climate of Xinchang tends to facilitate microbial activities, increasing soil methane absorption.

---

## [Decision Letter · Decision Letter 3]

1 Nov 2024

PONE-D-24-03564R3Characterization of methane microseepage from natural gas reservoirs in mild climate: a case study of Xinchang gas fieldPLOS ONE

Dear Dr. Wang,

Thank you for submitting your manuscript to PLOS ONE. After careful consideration, we feel that it has merit but does not fully meet PLOS ONE’s publication criteria as it currently stands. Therefore, we invite you to submit a revised version of the manuscript that addresses the points raised during the review process.

We look forward to receiving your revised manuscript.

Kind regards,

Trung Quang Nguyen

Academic Editor

PLOS ONE

Reviewers' comments:

Reviewer's Responses to Questions

**Comments to the Author**

1. If the authors have adequately addressed your comments raised in a previous round of review and you feel that this manuscript is now acceptable for publication, you may indicate that here to bypass the “Comments to the Author” section, enter your conflict of interest statement in the “Confidential to Editor” section, and submit your "Accept" recommendation.

Reviewer #1: All comments have been addressed

2. Is the manuscript technically sound, and do the data support the conclusions?

Reviewer #1: Yes

3. Has the statistical analysis been performed appropriately and rigorously? 

Reviewer #1: Yes

4. Have the authors made all data underlying the findings in their manuscript fully available?

Reviewer #1: Yes

5. Is the manuscript presented in an intelligible fashion and written in standard English?

Reviewer #1: Yes

6. Review Comments to the Author

**Reviewer #1:**  The manuscript titled "Characterization of methane microseepage from natural gas reservoirs in mild climate: a case study of Xinchang gas field" provides a thorough investigation into the contribution of methane microseepage to atmospheric methane, a critical greenhouse gas. The study employs a combination of soil sampling, methane flux monitoring, and geochemical analysis to explore methane migration patterns and origin tracing. The use of acid-hydrolyzed hydrocarbons (AHH) and altered carbonates (AC) to identify geochemical anomalies is well-established, and the methodology aligns with previous studies, adding to the body of knowledge on geological methane emissions in mild climates. However, the study could benefit from a more in-depth discussion of the limitations related to short-term flux measurements and the potential biases introduced by local climate conditions.

One of the study's strengths is its comprehensive use of isotopic analysis to distinguish thermogenic methane from biogenic sources, reinforcing the findings on methane origin. Nonetheless, the absence of long-term monitoring limits the study's ability to assess temporal variations in methane flux, which could be significant given the dynamic nature of methane emissions. Furthermore, while microbial activity's influence on methane flux is acknowledged, more detailed analysis of its role in the study area would enhance the discussion.

This study contributes valuable data to the understanding of methane microseepage in mild climates but would benefit from addressing the identified limitations and considering the potential for integrating remote sensing techniques to improve methane detection.

Questions:

1. How could long-term monitoring influence the understanding of methane flux variations in the Xinchang gas field?

2. What specific factors related to local climate conditions could introduce biases in short-term methane flux measurements?

3. How do microbial activities specifically impact methane flux measurements in mild climates like the Xinchang gas field?

4. Could integrating remote sensing methods with in-situ monitoring provide a more comprehensive picture of methane microseepage?

5. How do the methane flux measurements in the Xinchang gas field compare with those from regions with more extreme climatic conditions?

7. PLOS authors have the option to publish the peer review history of their article (what does this mean? ). If published, this will include your full peer review and any attached files.

**Do you want your identity to be public for this peer review?** For information about this choice, including consent withdrawal, please see our Privacy Policy .

Reviewer #1: No

---

## [Author Response · Author response to Decision Letter 4]

18 Nov 2024

Questions:

1. How could long-term monitoring influence the understanding of methane flux variations in the Xinchang gas field?

Response: long-term monitoring would reduce the potential biases introduced by shorter surveillance time and provide insights to understanding longer-term trend of methane flux.

This has been reflected in Section 3.5.

2. What specific factors related to local climate conditions could introduce biases in short-term methane flux measurements?

Response: As indicated in the discussion, factors that could affect microbial activity level, such as temperature and humidity, will affect short-term methane flux measurements. As revealed by Klusman et al. [11], in mild climatic condition, methane flux measured in relatively warmer and wetter seasons are lower than in relatively colder and drier seasons.

This has been reflected in Section 3.5.

3. How do microbial activities specifically impact methane flux measurements in mild climates like the Xinchang gas field?

Response: As revealed by Klusman et al. [11], in warm and humid condition, as such in the Xinchang gas field, the ensemble effect of the soil microbial community presents as a methane sink, as methanotrophic oxidation is dominant.

This explanation has been reiterated in Section 3.5.

4. Could integrating remote sensing methods with in-situ monitoring provide a more comprehensive picture of methane microseepage?

Response: Yes, it would provide information on the spatial distribution of methane microseepage in a more timely and holistic fashion.

Corresponding changes have been made in Section 3.5.

5. How do the methane flux measurements in the Xinchang gas field compare with those from regions with more extreme climatic conditions?

Response:

As stated in the manuscript, Xinchang gas field belongs to the category of the lowest methane flux. This is the combined effect of the caprock seal, deep reservoir, and soil methanotrophic oxidation. Other regions with more extreme climatic conditions, such as Alaska (cold and dry, polar), Venezuela (hot and humid, tropical), and Arabian Peninsula (hot and dry, desert), have all seen much higher methane fluxes.

However, note that it is impossible to draw any general conclusions by directly comparing locations of different climatic conditions for their methane fluxes. This is because climatic condition, by affecting soil methanotrophic oxidation, is only one of the many factors that may impact methane flux, whilst other major ones include geological setting and reservoir depletion level, all unique to each case of oil and gas field.

These changes have been reflected in the second last paragraph of Section 3.4.

---

## [Decision Letter · Decision Letter 4]

4 Dec 2024

Characterization of methane microseepage from natural gas reservoirs in mild climate: a case study of Xinchang gas field

PONE-D-24-03564R4

Dear Dr. Haoqi Wang,

We’re pleased to inform you that your manuscript has been judged scientifically suitable for publication and will be formally accepted for publication once it meets all outstanding technical requirements.

Kind regards,

Trung Quang Nguyen

Academic Editor

PLOS ONE

Additional Editor Comments (optional):

Reviewers' comments:

Reviewer's Responses to Questions

**Comments to the Author**

1. If the authors have adequately addressed your comments raised in a previous round of review and you feel that this manuscript is now acceptable for publication, you may indicate that here to bypass the “Comments to the Author” section, enter your conflict of interest statement in the “Confidential to Editor” section, and submit your "Accept" recommendation.

Reviewer #1: All comments have been addressed

2. Is the manuscript technically sound, and do the data support the conclusions?

Reviewer #1: Yes

3. Has the statistical analysis been performed appropriately and rigorously? 

Reviewer #1: Yes

4. Have the authors made all data underlying the findings in their manuscript fully available?

Reviewer #1: Yes

5. Is the manuscript presented in an intelligible fashion and written in standard English?

Reviewer #1: Yes

6. Review Comments to the Author

Reviewer #1: The authors have addressed the reviewers' comments comprehensively, significantly improving the manuscript's clarity, methodological rigor, and scientific contribution. The study provides a thorough investigation into methane microseepage from natural gas reservoirs, exemplified by the Xinchang gas field. The incorporation of new analyses and detailed discussions enhances the manuscript’s value.

First, the revised manuscript includes an expanded discussion on the influence of climate and microbial activity on methane flux measurements. This addition effectively contextualizes the study within broader climatic and biological frameworks, providing insights into the variability of methane emissions under different environmental conditions. Furthermore, the authors have clarified the methodological approach, particularly the integration of static gas chambers and isotopic analyses, which ensures the robustness of the methane flux measurements.

The response to comments regarding long-term monitoring and remote sensing integration demonstrates the authors' commitment to addressing methodological limitations. Their acknowledgment of spatial and temporal variability in methane flux and the potential for remote sensing to complement in-situ measurements reflects a well-rounded and forward-looking perspective.

Moreover, the comparison between Xinchang and other global sites enriches the discussion, highlighting the uniqueness of the observed patterns in mild climates and their implications for geological methane management. The authors also appropriately emphasize the need for continued research into the complex interactions of geological, climatic, and microbial factors influencing methane emissions.

The conclusions are well-supported by the results, particularly the identification of methane release patterns and the confirmation of thermogenic origins through isotopic analyses. These findings are not only relevant to the Xinchang gas field but also provide valuable insights for managing methane microseepage in similar geological settings globally.

Therefore, the authors have significantly improved the manuscript, addressing all major concerns and enhancing the scientific quality and clarity of their work. The study is now suitable for publication and will contribute meaningfully to the understanding and management of geological methane emissions

7. PLOS authors have the option to publish the peer review history of their article (what does this mean? ). If published, this will include your full peer review and any attached files.

**Do you want your identity to be public for this peer review?** For information about this choice, including consent withdrawal, please see our Privacy Policy .

Reviewer #1: No

---

## [Editor Report · Acceptance letter]

PONE-D-24-03564R4

PLOS ONE

Dear Dr. Wang,

I'm pleased to inform you that your manuscript has been deemed suitable for publication in PLOS ONE. Congratulations! Your manuscript is now being handed over to our production team.

Kind regards,

on behalf of

Dr. Trung Quang Nguyen

Academic Editor

PLOS ONE